# Thermal and Mechanical Degradation of Recycled Polylactic Acid Filaments for Three-Dimensional Printing Applications

**DOI:** 10.3390/polym14245385

**Published:** 2022-12-09

**Authors:** Dongoh Lee, Younghun Lee, Inwhan Kim, Kyungjun Hwang, Namsu Kim

**Affiliations:** 1Department of Mechanical Engineering, Konkuk University, Seoul 05029, Republic of Korea; 2Gangneung Science and Industry Promotion Agency, Gangneung 25440, Republic of Korea

**Keywords:** poly-lactic acid, 3D printing, thermal degradation, mechanical degradation, recycling, molecular weight, additive manufacturing

## Abstract

The recycling of filaments used in three-dimensional (3D) printing systems not only mitigates the environmental issues associated with conventional 3D printing approaches but also simultaneously reduces manufacturing costs. This study investigates the effects of successive recycling of polylactic acid (PLA) filaments, which were used in the printing process, on the mechanical properties of recycled filaments and printed objects. The mechanical strengths of the printed PLA and the adhesion strengths between 3D-printed beads were evaluated via the tensile testing of the horizontally and vertically fabricated specimens. Gel permeation chromatography analysis revealed a reduction in the molecular weight of the polymer as a result of recycling, leading to a decrease in the mechanical strength of the 3D-printed product. Additionally, scanning electron microscopy images of the cutting plane showed that the fabricated beads were broken in the case of the horizontally fabricated specimen, whereas in the case of the vertically fabricated samples, the adhesion between the beads was weak. These findings indicate that the mechanical strength in the in-plane and out-of-plane directions must be improved by increasing the mechanical strength of the bead itself as well as the adhesion strength of the beads.

## 1. Introduction

Additive manufacturing (AM) using three-dimensional (3D) printing is an emerging technology that offers considerable design freedom, particularly in the design of rapid prototypes, personalized medical components, and parts that cannot be mass-produced [1,2,3,4]. Similar to other AM techniques, components or systems used in 3D printing are fabricated via the deposition of successive layers, rather than the removal of material from a larger piece, as in subtractive manufacturing (e.g., a lathe) [5]. Among the currently available 3D printing technologies, fused deposition modeling (FDM), in particular, has been widely utilized in various applications owing to its low cost. In 3D printers that employ the FDM method, a thermoplastic filament is heated to temperatures close to its melting point, followed by layer-by-layer extrusion to create 3D objects. This technology has been employed in a wide range of applications; however, its increased use has raised several environmental concerns. Therefore, the production of filaments from eco-friendly recycled materials has been investigated to mitigate environmental issues. Recently, the recycling of polymer waste to alleviate the environmental issues caused by polymeric waste accumulation has received significant attention. Moreover, most polymeric materials are produced from oil and gas; therefore, the increased production of polymer-based materials ultimately increases the exploitation of natural resources [6].

Currently, various raw materials are used in FDM-type 3D printers, such as acrylonitrile butadiene styrene (ABS), nylon, polycarbonate, high-density polyethylene, high-impact polystyrene, and polylactic acid (PLA). Among these materials, PLA has received considerable attention owing to its relative abundance and cost-effectiveness as well as its relatively low melting point (150–160 °C), which reduces the energy required for printing. Additionally, PLA is an important biodegradable polyester that can be used in biomedical and pharmaceutical applications, such as implant devices, tissue scaffolds, and internal structures. Therefore, numerous studies have been conducted for characterizing 3D-printed PLA materials, such as fatigue, anisotropy, crystallinity, heat conduction, composites, dimensional accuracy, and dielectric property analyses [7,8,9,10,11,12,13,14]. Additionally, PLA is generally derived from renewable resources, such as starch and sugar, and is therefore a safer alternative to potentially toxic ABS plastics [6,15,16,17]. 

The increase in the use of filaments in 3D printing applications has led to an increased interest in recycling strategies, which can reduce the cost of feedstock, manufacturing cost of fabricated products, and generation of waste [18,19]. In addition, recycling reduces greenhouse gas emissions and lowers the environmental impact of products fabricated using 3D printers [18,19,20,21,22]. Several previous studies have examined the recycling process for 3D printing applications. Baechler et al. developed an extruder to produce filaments from polymer wastes, such as high-density polyethylene (HDPE) from waste bottles and laundry detergent containers [23]. McNaney reported on Filabot, which is a plastic filament producing company that utilizes post-consumer plastic waste, thereby reducing the manufacturing cost of 3D-printed models. Kim et al. designed a recycling system that includes a shredder that crushes the output of 3D printing and any generated waste as well as a spooler that enables the recycled filament to be used directly in the 3D printer [24,25]. Considering other thermoplastics, mechanical (or physical) recycling is more commonly used for PLA than chemical recycling and reuse. This process involves mechanically grinding the plastic into small pieces and subsequently reprocessing and compounding them at elevated temperatures to produce a new component or filament. This process is typically conducted using a recycling system; the steps involved in the recycling process are illustrated in Figure 1 [25]. However, thermal and mechanical degradation, similar to that observed during injection molding and extrusion, might occur as a result of this process [6,15,26,27,28]. Thus, understanding the degradation mechanism of recycled PLA filaments, as well as any object produced using them via FDM type 3D printing, is critical.

Several studies have been conducted to investigate the mechanical properties and degradation mechanisms of 3D-printed specimens and to eliminate the degradation caused by recycling. Lanzotti et al. reported the impact of process parameters, such as layer thickness, infill orientation, and the number of shell parameters, on the mechanical properties of 3D-printed PLA specimens [29]. These studies clarify the impact of process factors on the mechanical properties of 3D-printed specimens. To improve the thermal stability and mechanical properties of PLA, Yang et al. employed crosslinking via the chemical treatment of the melt by adding small amounts of triallyl isocyanurate and dicumyl peroxide as crosslinking agents [17]. The results showed a decrease in crystallinity and a significant improvement in the thermal stability, tensile modulus, and strength of PLA. In another study, the dopamine coating of the PLA pellets used in the 3D printing process, to improve the adhesion between layers fabricated by the 3D printer, was investigated [30]. The mechanical properties of the recycled specimen improved after polydopamine coating. Thus, to improve the thermal and mechanical properties of 3D-printed parts, understanding the degradation mechanism of materials and the main cause of failures is critical.

This study investigates the feasibility of recycling commercialized PLA filaments by employing a distributed recycling system which was designed and assembled in the laboratory. Furthermore, the mechanism of degradation resulting from the recycling process is presented. Compared to centralized recycling systems, distributed recycling is advantageous when the amount of waste materials for recycling is relatively small. Tensile tests were performed to compare the mechanical properties of the 3D-printed specimens fabricated using pristine and recycled PLA. Further, gel permeation chromatography (GPC) was performed at each recycling stage to better understand the degradation mechanism and measure any change in the molecular weight of the PLA, which is directly related to the mechanical strength of the polymer. In addition, the mechanical strengths of the horizontally and vertically fabricated specimens were also tested and compared. The fracture surfaces of the specimens were observed using scanning electron microscopy (SEM) to investigate the fracture mode.

## 2. Materials and Methods

The recycling of the PLA filament products was conducted using a custom mechanical recycling system consisting of a shredder, extruder, spooler, sensor, and controller, as shown in Figure 1 [24,25]. Using this approach, failed or broken parts fabricated using a 3D-printer were broken into small pieces using a shredder comprising an auger driven by an electric motor, followed by heating to temperatures close to the glass transition temperature for softening. The optimized recycler setting for extruding filament was summarized in the reference which was published by same authors [25]. The resulting product was forced through a die to extrude the PLA filament. The temperature of the heating zone and speed of the extruder were determined by recycled materials and controlled via a closed-loop controller to regulate the diameter of the extruded filament. The details of this process can be found another reference [26]. To investigate the mechanical performance of the recycled PLA filament, an open-source FDM type 3D printer (Cubicon 3DP-110F, Hyvision System Corp., Seongnam, Republic of Korea) with a 0.4 mm-diameter nozzle was used to fabricate Type-5 tensile test specimens, as shown in Figure 2, using a PLA filament with 1.75 mm-diameter, as per ASTM standard D638 for the tensile properties of plastics. To investigate the impact of recycling on the mechanical properties of PLA, the pristine material was compared to samples recycled once and three times.

Tensile testing was performed using a universal testing machine (Instron 5569, Instron Corp., Norwood, MA, USA) with a grip speed of 5 mm/min and a grip distance of 25.4 mm. The data of at least four samples for each test were used to calculate the average value to ensure consistency. All tensile tests were performed at 25.2 °C and 45.5% relative humidity using the two types of test specimens, as shown in Figure 2. The specimens fabricated horizontally (Figure 2a) were used to investigate the mechanical properties of the material itself, whereas those vertically produced (Figure 2b) were used to investigate the mechanical properties and adhesion between the fabricated beads. In both instances, the temperature of the nozzle was maintained at 210 °C, as per the recommendations of the material supplier. The heating bed and chamber were maintained at 65 °C and 45 °C, respectively. The air gap between the beads of the fabricated material was set to zero. The ultimate tensile strength was measured as the average value for at least four specimens and was used to investigate the effects of recycling on the initial mechanical properties of the material. The fracture surfaces produced via tensile testing were investigated using SEM at an accelerating voltage of 5 kV (S-4800, Hitachi High-Technologies Corp., Schaumburg, IL, USA). 

GPC analysis was performed using a WATERS GPC system with a Waters 410 differential refractometer and Shodex LF-804 (7.8 mm × 300 mm) column. As this type of size-exclusion chromatography separates analytes based on their size, it is a useful analytical tool for monitoring the molecular weight of a polymer. PLA was dissolved in chloroform, and the resulting mobile phase was passed through a column with a highly porous structure, where separation occurs based on the hydrodynamic volume (i.e., the radius of gyration) of each analyte. The molecular weights of pristine and recycled PLA were compared as they affect several characteristic physical properties of a polymer, such as its tensile strength, adhesive strength, brittleness, elastic modulus, and melt viscosity.

The differential scanning calorimetry(DSC) was carried out to investigate the change in the degree of crystallinity according to the number of recycling process(DSC Q20, TA Instrument, New Castle, DE, USA). A set of heating and cooling processes were carried out following three steps: PLA was heated to 200 °C at 5 °C/min with purged nitrogen environment to remove its thermal history. The PLA was then cooled down to 20 °C and heated back to 200 °C at the same rate. The degree of crystallinity(*X_c_*) was calculated by using the below equation based on the second heating:(1)$$X_{c}=\frac{\Delta H_{m}}{\Delta H_{0}}\times 100$$
where $\Delta H_{m}\;$ is the heat of melting and $\Delta H_{m}\;$ is the heat of meting for an infinitely large crystal, 93.6 g/J).

## 3. Results and Discussion

Printed and failed parts from 3D printing processes were used to extrude the recycled PLA filaments using a custom mechanical recycling system. The mechanical strength of recycled PLA specimens (which is an important parameter in 3D-printed parts) fabricated using recycled filaments was investigated via tensile tests. Figure 3 and Figure 4 show the tensile stress and strain curves of the pristine and recycled PLA specimens fabricated in the horizontal and vertical directions, respectively. Several specimens were broken near the grip location during the tensile test, and the data from these specimens were excluded. This phenomenon was more frequently observed in the specimens fabricated using recycled filaments, which can be attributed to the shredded specimen being pulled into the filament without fully mixing during the recycling process. Among the two specimens, the ultimate tensile strength of the horizontally fabricated specimen was higher than that of the vertically fabricated specimen. This indicates that the strength of the fabricated PLA material is higher than the adhesion strength between the fabricated beads, resulting in the typical anisotropic property of the FDM-type 3D-printed product. These findings are consistent with those from previous studies [31,32,33,34]. In addition, in the case of the vertically fabricated pristine PLA specimen, no elongation was observed after the maximum stress before breaking, which implies that the fracture occurs between the adhesion interfaces and not in the material itself.

Considering the change in the mechanical strength of the horizontally fabricated specimens after recycling, both the maximum stress before break and the strain at break of the horizontally fabricated specimens decreased as the number of recycling steps increased, as shown in Figure 3 and Figure 5. After three recycling steps, the maximum tensile strength and strain at break of the horizontally fabricated specimens reduced by 38.7% and 26.3%, respectively, on average. The specimens for the tensile strength test were also prepared by recycling the material up to five times; however, the tensile strength data of these samples were excluded owing to the large deviations in the measured values for these samples. The experimental results indicate that recycling led to a decrease in the mechanical strength and an increase in the brittleness of the material. The degradation of the mechanical strength of the polymeric material is due to a reduction in its molecular weight [4,27,34,35,36,37]. This is consistent with previous reports suggesting that mechanical recycling at elevated temperatures degrades the macromolecular structure, resulting in the chain scission of the polymer structure [4,27,35,36,37]. The resulting shorter chains increase the number of chain ends in the structure and the stress at which fracture occurs [38]. Similar observations regarding the degradation of PLA have shown that a correlation generally exists between the tensile strength and molecular weight of polymer materials, which can be approximated by the inverse relation [37,39,40]:(2)$$S=S_{\infty }-\frac{A}{M}$$
where *S_∞_* is the saturated tensile strength for an infinite molecular weight, and *A* is the correction factor for the material type.

To determine the change in the molecular weight of PLA caused by recycling, the weight-averaged molecular weights of pristine and recycled PLA were determined using GPC with polystyrene as reference material. In Figure 6, the shift in the peak from left to right indicates a longer elution time, and therefore, a lower molecular weight. This elution time was correlated with that of the reference material to obtain the molecular weights, as shown in Figure 7, revealing that the molecular weight of the pristine material (175,888 g/mol) was reduced to 90,021 g/mol after five recycles. Recycling at elevated temperatures and repeated 3D printing processes are therefore considered conducive to chain scission, resulting in the degradation of the mechanical properties of PLA. However, other degradation mechanisms might also occur during recycling, such as the depolymerization of macromolecular chains due to residual catalyst, non-radical reactions, and mechanical degradation due to interactions between PLA and the equipment used in the fabrication process [4,41].

The degree of crystallinity was calculated using Equation (1), and results were summarized in Table 1. It was found that the degree of crystallization increased with the number of recycling. Based on previous studies, the change in crystallinity of polymer according to the degradation of materials can be explained in two different ways. The first one is that the degree of crystallinity decreased with increased number of recycling process. It was reported that the decrease in crystallinity is due to the radical reactions [42]. The radical reactions generated after recycling might cause the crosslinking of the polymer and reduce its crystallinity [43]. The second one is that the degree of crystallinity increased with the number of recycling of polymer. It was also reported that the increase in the degree of crystallinity with recycling is a kinetic effect due to the reduction of the molecular weight [15]. As provided in the previous paragraph of this work, the molecular weight of PLA decreases with recycling due to chain scissions. The shortened chain length due to chain scissions allows to have a better mobility resulting in a crystallite thickening process rather than new crystallization [44]. Therefore, it is expected that increased crystallinity induces the brittleness of PLA and resulting in the degradation of mechanical strength. In our study, it was confirmed that crystallinity increases with the number of recycling based on the results from GPC, DSC, and tensile testing experiments.

The maximum tensile strength and strain before breaking of the vertically fabricated PLA specimens also decreased with the number of recycling steps, as shown in Figure 4 and Figure 5. After three cycles, the maximum tensile strength and strain at break of the horizontally fabricated specimens reduced to 42.5% and 34.2%, respectively, on average. Therefore, the deterioration rate of the sample fabricated in the vertical direction is higher than that of the sample fabricated in the horizontal direction. This indicates that the chain scission of the PLA matrix reduces not only the strength of the material but also the adhesion between the beads of the fabricated PLA, which in the case of a 3D-printed structure, acts as an adhesive material. Thus, heating during recycling and the 3D printing process causes thermally induced chain scission and a reduction in cohesive strength. These results suggest that the maximum tensile strength of the vertically fabricated specimens can be improved by adding an adhesion promoter. Studies on improving the mechanical properties of polymers by coating them with an adhesion promoter, such as polydopamine (which is an adhesive polymer derived from mussels), have been conducted. Based on the results of this study on the degradation mechanism, the mechanical properties of the recycled filament were noticeably improved [30]. The maximum tensile strength of the vertically fabricated pristine PLA was lower than that of the horizontally fabricated specimen because the strength of a vertically fabricated specimen is dependent on the adhesion between the beads rather than the strength of the PLA beads. This is in agreement with previous results [45] and indicates that vertically fabricated specimens exhibit a more brittle behavior than horizontally fabricated specimens, that is, they break abruptly when stress is induced during a tensile test; this behavior is not exhibited by the pristine material. In contrast, horizontally fabricated specimens exhibited elongation even after reaching their ultimate tensile strength, which is consistent with the intrinsic properties of the polymer (particularly for the samples produced via injection molding). However, this elongation after the ultimate tensile strength decreased as the number of recycling steps increased, owing to the embrittlement of recycled PLA, as shown in Figure 3.

The cross-sectional area after tensile testing was investigated via SEM imaging, and the fracture surfaces of the two different specimens are presented in Figure 8. These figures clearly show the difference in fracture morphology, and the characteristic fracture surface of the horizontally fabricated specimen indicates that the fabricated PLA beads are broken. In contrast, Figure 8b reveals that the cracks propagate entirely through the bonding interface between the beads in the vertically fabricated specimen. This confirms that the mechanical strength of the PLA beads determines the mechanical strength of the horizontally fabricated specimens, whereas the adhesion strength between the beads governs the mechanical properties of the vertically fabricated specimens. As the former is stronger than the latter, this imparts anisotropic mechanical properties to the 3D-printed objects. Hence, these factors must be considered during the design and manufacturing of products via 3D printing. As the surface morphologies of the fracture surfaces are closely correlated with the mechanical characteristics of printed parts, lots of research have been reported on the method of characterizing the surface topologies of the fracture surface such as optical microscope, scanning electron microscopy, X-ray computed tomography and the correlation with the mechanical properties [42,43,44,45,46]. Based on results in the previous reports, it could be confirmed that, as the void in the fracture surface decreases, the mechanical strength of the printed parts increases. Although not included in this study, it would be an important study to optimize the process conditions to reduce void and improve the mechanical properties when using recycled filaments [46]. This part was not included in this study and was left as a topic for the next research.

## 4. Conclusions

The recycling of polymeric materials enables waste materials to be converted into filaments, which can then be used repeatedly in 3D printing systems. PLA, which has been widely used in 3D printing applications, was recycled via a distributed recycling system, and the recycled PLA filaments and printed specimens were characterized to investigate the effectiveness of the recycled filament. The characterization of the mechanical properties of the PLA filament, for use in 3D printers, reveals that mechanical recycling induces thermally activated degradation, thereby reducing the molecular weight of the polymer with each successive cycle. This results in the chain scission of PLA and degradation of the mechanical properties of the fabricated product. Using two different types of specimens (horizontally and vertically fabricated), mechanical recycling was shown to degrade the mechanical strength of the PLA bead as well as the adhesion between the beads. Thus, the mechanical strength of the PLA filaments was sensitive to the number of successive thermomechanical recycling processes. Therefore, recycled filaments are more suitable for fabricating models and prototypes than the parts operating under heavy loads because of the degradation of their mechanical strength. The investigation of the degradation mechanism revealed that this limitation can be overcome using chain extenders or adhesion promoters during recycling.

## Figures and Tables

**Figure 1 polymers-14-05385-f001:**
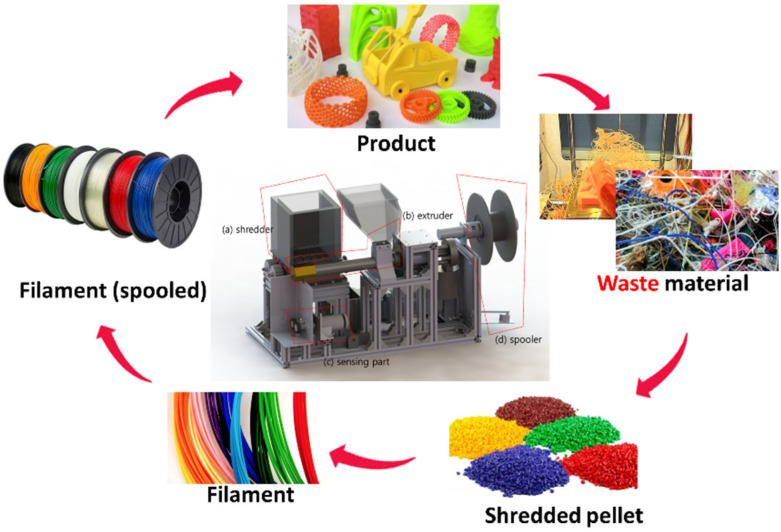
Steps involved in the recycling process of initial waste materials to form filaments (outside view) and an inside schematic of the filament recycling system, consisting of a shredder, extruder, sensor, and spooling systems.

**Figure 2 polymers-14-05385-f002:**
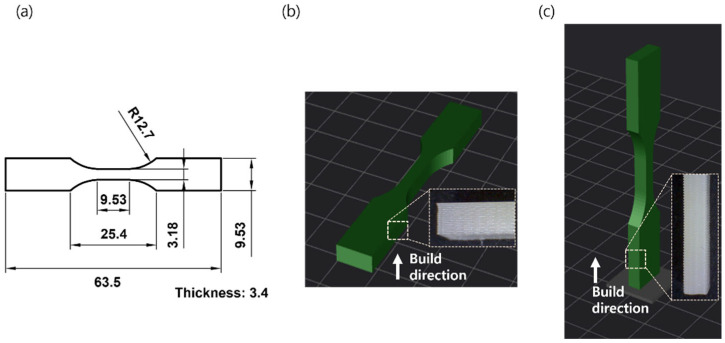
(**a**) Graphical representation of the tensile test specimen specified in the ASTM standard D638-10 (type 5); tensile test specimens printed (**b**) horizontally and (**c**) vertically. Magnified images of the specimens are also shown.

**Figure 3 polymers-14-05385-f003:**
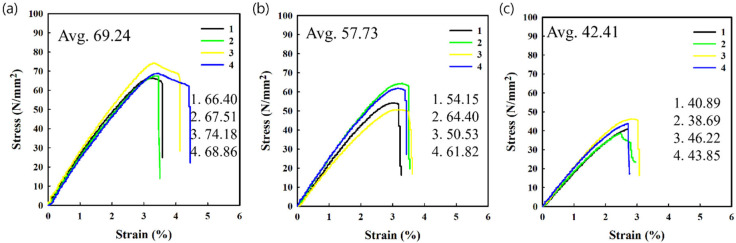
Stress–strain curves for (**a**) pristine, (**b**) one-time recycled, and (**c**) three-time recycled specimens fabricated horizontally.

**Figure 4 polymers-14-05385-f004:**
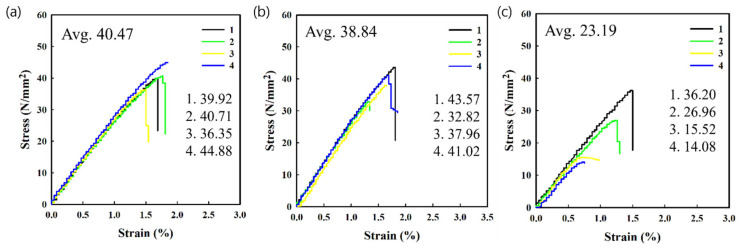
Stress–strain curves for (**a**) pristine, (**b**) one-time recycled, and (**c**) three-time recycled specimens fabricated vertically.

**Figure 5 polymers-14-05385-f005:**
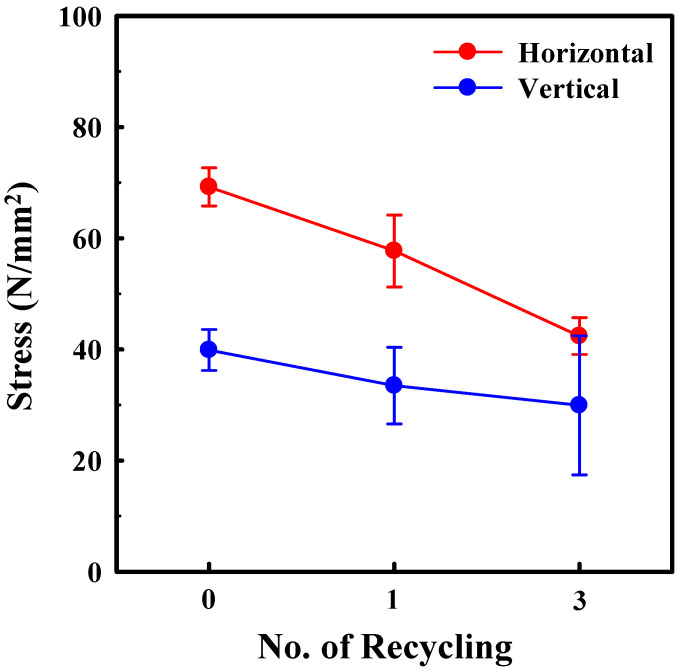
Averaged ultimate tensile strength as a function of the number of recycling steps for both horizontally and vertically fabricated specimens.

**Figure 6 polymers-14-05385-f006:**
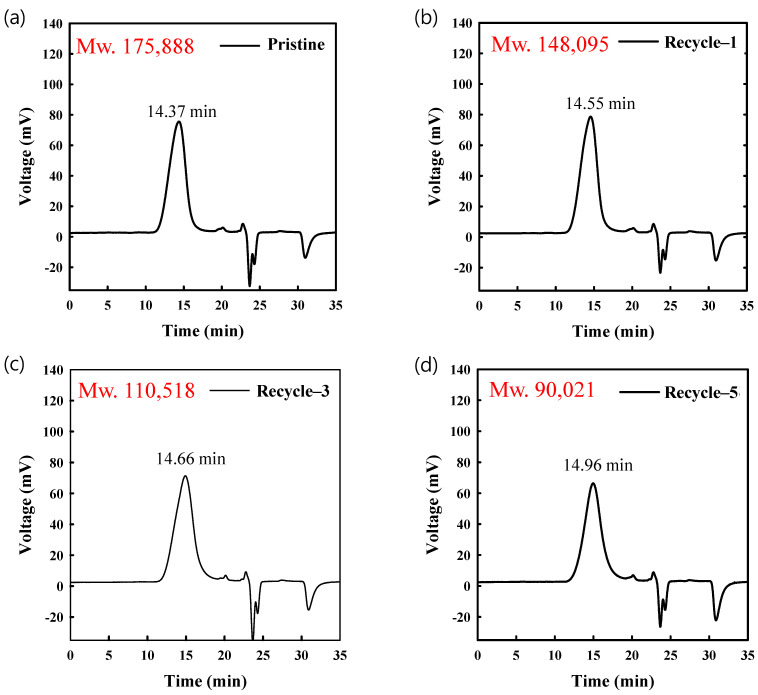
GPC results for pristine and recycled PLA filament.

**Figure 7 polymers-14-05385-f007:**
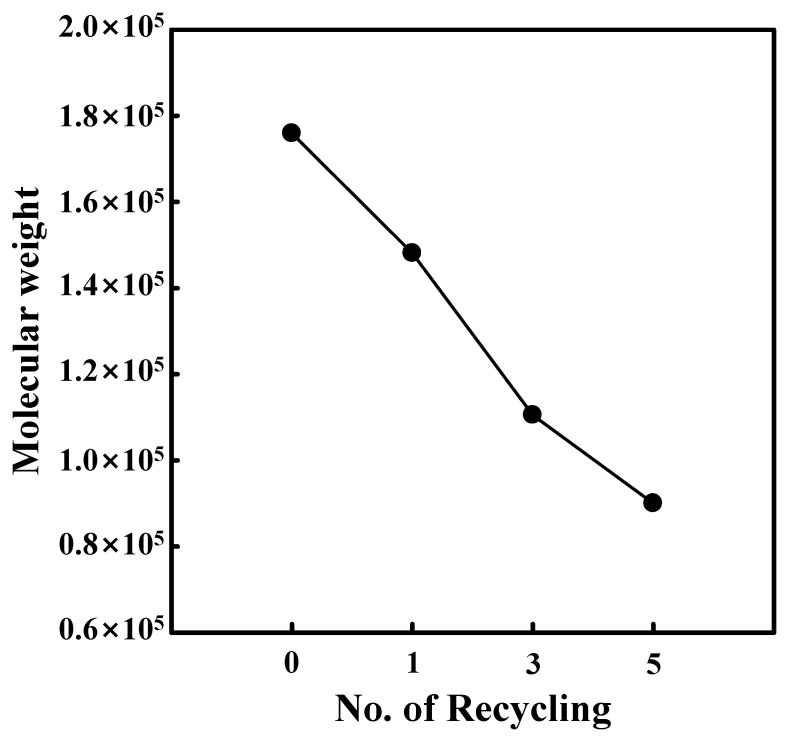
The molecular weight of PLA as a function of the number of recycling steps.

**Figure 8 polymers-14-05385-f008:**
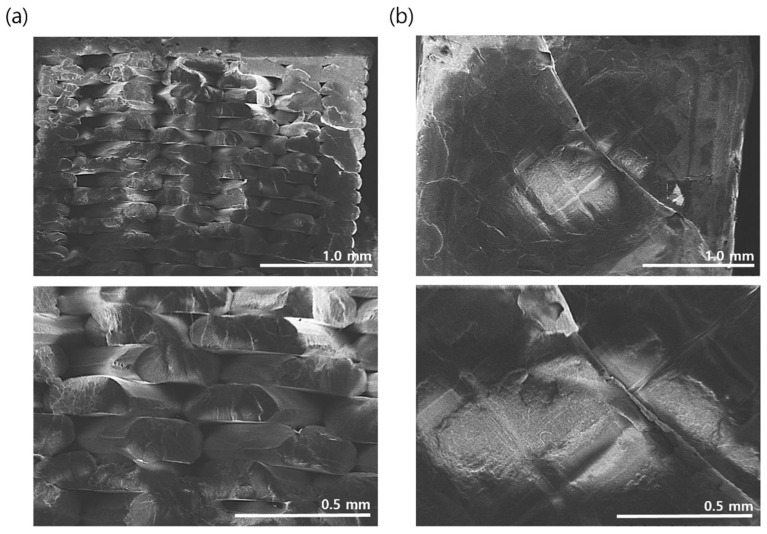
Fracture surface morphology of 3D-printed specimens after the tensile test: (**a**) horizontally and (**b**) vertically fabricated specimens.

**Table 1 polymers-14-05385-t001:** Evaluating the thermal properties of PLA as a function of the recycling process.

Sample (PLA)	∆*H_m_*	∆*H*_0_ (J/g)	*Xc* (%)
Pristine	30.89	93.6	33.0
Recycle 1	33.99	93.6	36.3
Recycle 3	38.66	93.6	41.3
Recycle 5	38.09	93.6	40.7

## Data Availability

Not applicable.

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
