# Peer review of "Thermal and Mechanical Degradation of Recycled Polylactic Acid Filaments for Three-Dimensional Printing Applications"

_polymers, 2022, doi:10.3390/polym14245385_

Round 1
Reviewer 1 Report
The manuscript reports a study on the effect of recycling steps on the performances of 3d printed PLA parts. The authors assess a loss in molecular weight and on mechanical properties of the material after each recycling step.
In this reviewer's opinion, the work does not contain enough elements of novelty and enough amount of data to deserve publication in Polymers.
The authors must analyze the effects of different printing and recycling conditions, and try to assess the effects of each processing step: printing, shredding, extrusion, and new printing.
Some secondary points:
- the full conditions of each step of processing must be provided, including an estimation of the residence time at high temperature, which heavily affect the degradation of PLA
- it is not clear if the material was dried or not before each of the melting steps
Author Response
We deeply appreciate the time and efforts of editor and referees in reviewing our manuscript and valuable comments and suggestions. We have addressed all comments from the reviewers as outlined below. The manuscript was significantly modified according to the suggestions of reviewers. We hope that the manuscript is now suitable for publication.
File for response to reviewer's comment has been attached.

Reviewer 2 Report
In this article, authors studied the thermal and mechanical degradation of recycled polylactic acid filaments for three-dimensional printing applications. The only analysis related to what exactly happens to polymer itself was presented with GPC, which clearly shows significant decrease in molecular weight during mechanical recycling. Authors should explore also thermo-mechanical properties of recycled polymer itself, without printing, as a standard reference before comparing the end results with printed polymers. Also some of test results suggest that the differences are not really significant considering the standard deviation among measured samples. This should also be discussed in detail. Overall, it is an interesting topic for broad 3D printing community, but authors need to run couple of control experiments before it can be recommended for publication.
Author Response

(The authors gave the same response as above.)

Reviewer 3 Report
In this manuscript, thermal and mechanical degradation of recycled polylactic acid filaments were studied in 3D printing applications using tensile tests, GPC and SEM analyses. Overall, the manuscript is well written and interesting results are reported.
Some minor observations are:
-Several references are missing. eg. line 83: Lanzotti et al. reported…, line 88: Yang et al. employed…, line 95…
-Materials and method section: What was the grade and supplier of PLA filament?
-Line 138: Why the relative humidity was only 45%? Typically 50% is used
-Line 194-196: Have the authors studied the effect of recycling steps on crystallinity of PLA?
-Figure 8: Can you specify the samples in this figure? Are these pristine material or recycled?
-The manuscript stated that mechanical recycling reduces mechanical strength and decreases molecular weight. Have the authors studied how much recycled PLA could be mixed with pristine PLA without losing the properties of the material?
Author Response

(The authors gave the same response as above.)

Round 2
Reviewer 1 Report
The manuscript marginally changed with respect to the previous version, thus the comments remain the same.
Author Response
Even though we have not significantly modified our manuscript, we have attached file for "Response to reviewer's comments" to address a point-by-point response to reviewer's comments.

Reviewer 2 Report
Authors have significantly revised the manuscript according to reviewer`s comment. I still suggest minor revision so that authors can discuss more clearly the fractured morphology as it is inherent in 3D printing that voids and defects are visible. Mechanical properties are thus the function of the morphology and please discuss accordingly. Refer to several existing literatures including: https://doi.org/10.3390/polym14091838 which shows the role morphology in mechanical properties for printed PLA.
Author Response
We deeply appreciate reviewer's comments for our work. As suggested by reviewer, we added more explanation about topologies of fracture surface including references. For your reference, we have attached file for modified parts.
